# Immunodeficiencies Push Readmissions in Malignant Tumor Patients: A Retrospective Cohort Study Based on the Nationwide Readmission Database

**DOI:** 10.3390/cancers15010088

**Published:** 2022-12-23

**Authors:** Wenchen Wang, Qingyu Meng, Yiping Cheng, Yalin Han, Yonggan Xue, Yanshen Kuang, Xuning Wang, Bobin Ning, Mu Ke, Zhipeng Teng, Sen Li, Peng Li, Hongyi Liu, Xiude Fan, Baoqing Jia

**Affiliations:** 1School of Medicine, Nankai University, Tianjin 300071, China; 2Department of General Surgery, The First Medical Centre, Chinese PLA General Hospital, Beijing 100853, China; 3Department of Endocrinology, Shandong Provincial Hospital, Shandong First Medical University, Jinan 250021, China; 4The Air Force Hospital of Northern Theater PLA, Shenyang 110016, China

**Keywords:** immunodeficiency, malignant tumor, readmission, post discharge decision

## Abstract

**Simple Summary:**

In this study, our primary aim was to evaluate the association between immunodeficiency diseases and the short- and long-term readmission risk for the 16 most common malignant tumors. A total of 603,831 patients diagnosed with malignant tumors at the time of index hospitalization were ultimately selected from the Nationwide Readmissions Database (NRD) of 2018 to establish 30-day, 90-day and 180-day readmission cohorts, respectively. We found that immunodeficiencies were independently associated with higher readmission risks for colorectal cancer, lung cancer, NHL, prostate cancer or stomach cancer. In order to detect patients at a high risk of early readmission and to reduce the burden on society, strategies should be considered to prevent the causes of readmission as a post discharge plan in this population.

**Abstract:**

Background: Immunodeficiency diseases (IDDs) are associated with an increased proportion of cancer-related morbidity. However, the relationship between IDDs and malignancy readmissions has not been well described. Understanding this relationship could help us to develop a more reasonable discharge plan in the special tumor population. Methods: Using the Nationwide Readmissions Database, we established a retrospective cohort study that included patients with the 16 most common malignancies, and we defined two groups: non-immunodeficiency diseases (NOIDDs) and IDDs. Results: To identify whether the presence or absence of IDDs was associated with readmission, we identified 603,831 patients with malignancies at their time of readmission in which 0.8% had IDDs and in which readmission occurred in 47.3%. Compared with NOIDDs, patients with IDDs had a higher risk of 30-day (hazard ratio (HR) of 1.32; 95% CI of 1.25–1.40), 90-day (HR of 1.27; 95% CI of 1.21–1.34) and 180-day readmission (HR of 1.28; 95% CI of 1.22–1.35). More than one third (37.9%) of patients with IDDs had readmissions that occurred within 30 days and most (82.4%) of them were UPRs. An IDD was an independent risk factor for readmission in patients with colorectal cancer (HR of 1.32; 95% CI of 1.01–1.72), lung cancer (HR of 1.23; 95% CI of 1.02–1.48), non-Hodgkin’s lymphoma (NHL) (HR of 1.16; 95% CI of 1.04–1.28), prostate cancer (HR of 1.45; 95% CI of 1.07–1.96) or stomach cancer (HR of 2.34; 95% CI of 1.33–4.14). Anemia (44.2%), bacterial infections (28.6%) and pneumonia (13.9%) were the 30-day UPR causes in these populations. (4) Conclusions: IDDs were independently associated with higher readmission risks for some malignant tumors. Strategies should be considered to prevent the causes of readmission as a post discharge plan.

## 1. Introduction

Cancer is a collection of diseases characterized by abnormal and uncontrolled cell growth that are mainly caused by genetic mutations [1,2]. Genetic mutations cause tumors to produce more highly specific neoantigens called tumor-specific antigens (TSAs), which are different from tumor-associated antigens (TAAs) [3,4]. The human immune system can be activated with the identification of TSAs, and T-cell-mediated cytotoxicity is able to reject cancer cells [3,5]. The TSAs present in most malignancies are associated with prompt responses to immune checkpoint inhibitors (ICIs), neoantigen vaccines and adoptive T-cell-receptor gene therapy [5,6,7]. Besides, both tumor-associated neutrophils (TANs) and tumor-associated macrophages (TAMs) play a dual potential role in the tumor microenvironment (TME) [8,9]. Natural killer (NK) cells, cytotoxic innate-like lymphocytes, are capable of recognizing cancer cells in order to mediate the spontaneous killing of cancer cells [10]. Moreover, there is growing evidence that both innate and adaptive immunity can contribute to the long-term clinical benefits of anticancer therapies such as chemotherapy and radiation therapy [11,12,13]. Therefore, a properly functioning immune system is vital to the prognosis of cancer.

Before the diagnosis of cancer, a small percentage of patients can be diagnosed with immunodeficiency diseases (IDDs) containing primary or secondary immunodeficiencies. In the general population, primary immunodeficiencies are quite rare in incidence, and the prevalence can range from 1/500 to 1/500,000 [14] of which 30% had common variable immunodeficiency (CVID) in a European internet-based database [15]. Of note, previous studies indicated that primary immunodeficiencies have a high risk of solid tumors and hematological malignancies [16,17]. The most common malignancies were non-Hodgkin lymphoma and gastric cancer in patients with CVID [16,17]. The major secondary immunodeficiency is human immunodeficiency virus (HIV). The number of aging people living with HIV that are on combination antiretroviral therapy is increasing, and their proportion among the total of people with HIV is estimated to increase from 28% in 2010 to 73% in 2030 [18]. Non-AIDS-defining malignancies, such as lung cancer, anal cancer and hepatocellular cancer, are the leading cause of death in the HIV population of highly developed countries [19,20,21]. In addition, the burden of IDDs and the proportion of cancer-related morbidity and mortality have ascended in IDD patients [14,16,17,19,21].

Considering the significant morbidity and mortality associated with IDDs in cancer patients, it is necessary to diagnose and treat those with IDDs. However, there have been limited population data identifying both the temporality and risk factors associated with post discharge in this population, which may be useful for both physicians and hospital administrators to guide the necessary resources and to direct readmission reduction programs.

However, no study has yet systematically examined the association between IDDs and the readmission risk for various types of cancers (lung cancer, colorectal cancer, stomach cancer, etc.). In this study, we aimed to determine whether the presence or absence of IDDs in cancer patients was associated with readmission, and we assessed the temporality and the causes of readmission among malignant tumor patients by using data from the Nationwide Readmissions Database (NRD).

## 2. Materials and Methods

### 2.1. Data Source

NRD, part of the Healthcare Cost and Utilization Project (HCUP), supports various types of analyses of national readmission rates of all patients, regardless of the expected payer for a hospital stay in the 28 states in an individual year. Based on the NRD unique record identifier, it allows one to track patients between hospitals across states but not across years. According to U.S. Census Bureau data, the 2018 NRD accounted for 59.7% percent of the U.S. population’s information on hospital readmissions for all ages and contained a full year of the International Classification of Diseases, Tenth Revision, Clinical Modification (ICD-10-CM) codes [22].

### 2.2. Study Population

We conducted a retrospective cohort study using a total of 12,928,231 patients that were admitted to the hospital in 2018 from the NRD. According to the ICD-10-CM codes, we identified patients with the 16 most common malignant tumors, which included bladder tumors, brain and other nervous tumors, breast tumors, cervical tumors, colorectal tumors, esophageal tumors, leukemia, liver tumors, lung tumors, non-Hodgkin lymphoma (NHL), ovarian tumors, pancreatic and prostate tumors, stomach tumors, thyroid tumors and uterine tumors [23]. Based on the status of their immune system, the cancer patients were classified into two groups: non-immunodeficiency diseases (NOIDD) and IDDs. IDD group contained 46 primary immunodeficiencies, according to the most recently updated classification of primary immunodeficiencies [24,25], and secondary immunodeficiencies, mainly including human immunodeficiency virus (HIV) infections (Appendix A). We excluded the following patients: (1) patients aged <18, (2) pregnant patients, (3) patients with autoimmune diseases, (4) patients that died during their first hospitalization, (5) patients with missing data on their baseline characteristics and (6) patients lost to follow-up. If a patient had multiple readmissions following index hospitalization, only the first readmission was taken into account. The details of the inclusion and exclusion criteria are given in Figure 1.

### 2.3. Outcomes

The primary outcome of the study was to assess the association between IDDs and the readmission risk for 16 types of cancer. The secondary outcome was to ensure which stage had the most readmissions at 180-day follow-up and to evaluate the causes of readmission at this stage.

### 2.4. Data Collection

The NRD contains characteristics regarding patient and hospital factors that were collected and analyzed in this study during admission. We identified patients’ demographic factors, including sex, age, hypertension, diabetes, hyperlipidemia, body mass index (BMI), mortality risk, illness severity, patient location, primary expected payer, median household income and readmission due to common malignant tumors. Hospital-based characteristics that were captured included length of stay (LOS), total charges and disposition of patient.

### 2.5. Statistical Analysis

Patient and hospital information were weighted to reflect nationally representative results based upon conversion factors provided by the NRD [22]. The baseline characteristics of malignant tumor patients with or without IDDs were described. Continuous variables were evaluated with the independent samples t test and were denoted as mean and standard deviation, while categorical variables were evaluated with the chi-square test and were denoted as frequency counts and percentages.

30-day, 90-day and 180-day readmission of patients were analyzed using multivariable Cox regression model adjusted for age, sex, BMI, hypertension, diabetes, hyperlipidemia, LOS, location of residence, insurance type and household income.

To evaluate the temporality and the category of readmissions with the chi-square test, the time of readmission was stratified in 30-day intervals up to 180 days and was set as a categorical variable. 

The common malignancies of 30-day UPRs were analyzed with the chi-square test, and risk of 30-day UPRs was estimated with multivariable Cox regression adjusted the same as above.

To assess the difference in the cause of 30-day readmission between the IDD and NOID patients, chi-square test was used. 

Two-sided tests were used in all hypothesis tests with a significance level of *p* < 0.05. All analyses were performed with SPSS 25.0 (SPSS, Chicago, IL, USA).

## 3. Results

### 3.1. Baseline Characteristics

We identified 603,831 patients with malignancies at index admission in 2018 from the NRD of which 4852 patients were diagnosed with IDDs. Men were more likely to be observed in the IDD cohort (61.7% vs. 50.9%, *p <* 0.001), and the hospitalization age of the IDD cohort tended to be young (18–45 years: 12.3% vs. 5.8%, *p <* 0.001) (45–60 years: 25.4% vs. 21.0%, *p <* 0.001). Basic disorders such as hypertension (54.0% vs. 60.9%, *p <* 0.001), diabetes (25.4% vs. 27.5%, *p* = 0.001) and hyperlipidemia (31.8% vs. 37.2%, *p <* 0.001) were less likely to occur in the IDD cohort, but the severity of illness (80.0% vs. 53.8%, *p <* 0.001) and the risk of mortality (61.0% vs. 47.5%, *p <* 0.001) seemed more serious. Compared with patients with NOIDDs, those in the IDD group had more severe LOSs (22.9% vs. 13.6%, *p <* 0.001) and higher hospitalization costs (USD 114,912 vs. USD 81,344, *p <* 0.001). IDD patients were more likely to live in “Central” counties (32.6% vs. 28.4%, *p <* 0.001), had low (0–25th percentile) household incomes (25.2% vs. 23.7%, *p <* 0.001) and accepted more Medicaid from the government (12.7% vs. 8.8%, *p <* 0.001). In addition, a higher readmission rate was observed in the IDD cohort (47.3% vs. 36.6%, *p <* 0.001). (This information is shown in Table 1.)

### 3.2. Readmission Risk of Malignancy for Patients Diagnosed with or without IDDs

Of all the malignant tumor patients, 93.3%, 79.4% and 57.4% were 30-day follow-up, 90-day follow-up and 180-day follow-up admissions, respectively (Figure 1). Compared with those with NOIDDs, the tumor patients with IDDs had a higher readmission risk at 30 days (HR of 1.32; 95% CI of 1.25–1.40), 90 days (HR of 1.27; 95% CI of 1.21–1.34) and 180 days (HR of 1.28; 95% CI of 1.22–1.35). Categorized according to real readmission, we defined planned readmission (PR) and UPR. The same results were observed in the IDD cohort for the 30-day (HR of 1.29; 95% CI of 1.22–1.38), 90-day (HR of 1.27; 95% CI of 1.21–1.34) and 180-day (HR of 1.29; 95% CI of 1.22–1.37) UPR risk. In addition, the PR cohort had a higher readmission risk than the UPR cohort in these periods (Table 2).

### 3.3. The Temporality and the Category of Readmissions

The readmission risk for malignancy patients with IDDs was highest in the first 30 days after discharge and was higher than that of NOIDD patients (37.9% vs. 26.2%, *p <* 0.001) (Figure 2A). In the other 30-day intervals, the readmission rates of IDD patients were also higher than those of NOIDD patients (Figure 2A). UPRs accounted for 82.4% of patients in the 30-day readmission group of the IDD cohort, and the number of readmissions decreased in each subsequent 30-day period (Figure 2B).

### 3.4. Subgroup Analysis of 30-Day Unplanned Readmission

We focused on patients who had 30-day UPRs and conducted a univariate analysis in the tumor subgroup. The results indicated that IDD patients with colorectal cancer, lung cancer, NHL, prostate cancer or stomach cancer had a higher risk of 30-day UPR compared with NOIDD patients (Figure 3). The further multivariable Cox regression analysis showed that IDDs were the 30-day UPR risk factors in patients with colorectal cancer (HR of 1.32; 95% CI of 1.01–1.72), lung cancer (HR of 1.23; 95% CI of 1.02–1.48), NHL (HR of 1.16; 95% CI of 1.04–1.28), prostate cancer (HR of 1.45; 95% CI of 1.07–1.96) or stomach cancer (HR of 2.34; 95% CI of 1.33–4.14). Additionally, both males (HR of 1.38; 95% CI of 1.28–1.49) and females (HR of 1.14; 95% CI of 1.02–1.27); all ages, including 18–45 years old (HR of 1.48; 95% CI of 1.26–1.74), 45–60 years old (HR of 1.45; 95% CI of 1.29–1.63) and more than 61 years old (HR of 1.15; 95% CI of 1.06–1.26); those with an LOS ≤ 10 days (HR of 1.27; 95% CI of 1.17–1.37); and those with an LOS > 10 days (HR of 1.34; 95% CI of 1.20–1.50) had a higher risk of 30-day UPR in the IDD group. However, as for patients with ovarian cancer, IDDs seemed to be a protective factor (HR of 0.44; 95% CI of 0.20–0.99) (Figure 4).

### 3.5. The Causes of 30-Day Unplanned Readmissions

A total of 611 IDD patients and 50,722 NOIDD patients were readmitted within 30 days which were diagnosed with colorectal cancer, lung cancer, NHL, prostate cancer or stomach cancer. The eight top tumor- and nontumor-related causes of 30-day UPRs in tumor patients with or without IDDs were listed in Figure 5A,B. The common causes of 30-day UPRs in tumor patients with IDDs were anemia (44.2%), fluid and electrolyte disorders (40.8%), infections (28.6%), secondary malignancies (25.2%), kidney failure (21.1%), pneumonia (13.9%), heart diseases (10.8%) and gastroesophageal reflux disease (GERD) (8.0%). Compared with those with NOIDDs, there were higher incidences of anemia (44.2% vs. 32.7%, *p <* 0.001), infections (28.6% vs. 17.9%, *p <* 0.001) and pneumonia (13.9% vs. 10.6%, *p* = 0.009) and lower incidences of secondary malignancies (25.2% vs. 50.5%, *p <* 0.001), heart diseases (10.8% vs. 18.4%, *p <* 0.001) and GERD (8.0% vs. 11.6%, *p <* 0.006) within 30 days of IDD patients’ discharge (Table 3).

## 4. Discussion

Based on this nationally representative longitudinal study, we reported that IDDs were an independent risk factor for the readmission, especially the 30-day UPR, of patients with malignancies, including colorectal cancer, lung cancer, NHL, prostate cancer or stomach cancer. Additionally, we found that the first 30 days after discharge had the highest readmission rate in malignancy patients with IDDs and that UPRs accounted for 82.4% of 30-day readmissions in the IDD group. Of note, the main nontumor causes of 30-day UPRs were anemia, fluid and electrolyte disorders, infections, secondary malignancies, kidney failure, pneumonia, heart diseases and GERD, which should be prevented after discharge in malignancy patients with IDDs. Overall, this was the first large-scale retrospective cohort study to systematically assess the impact of immunodeficiencies on the short- and long-term clinical outcomes of patients with malignant tumors, which allowed for the calibration of readmission risk stratification in these populations. 

Compared with the NOIDD patients, IDD patients with the 16 most common malignant tumors were more likely to be males and tended to be younger. A recent Czech nationwide study showed that the average age of the experience of the first CVID-related symptoms and that at the time of CVID diagnosis with malignant tumors were 34.2 and 38.3 years, respectively [17]. In a long-term cohort study regarding Asian HIV-infected patients with non-AIDS-defining cancers (NADC), half of them were aged 40–59 years and had an advanced-stage disease at diagnosis [26]. The above results suggested that the age of onset of cancer tended to be younger in IDD patients. 

Although there was a lower amount of underlying diseases in malignancy patients with IDD, we confirmed that these patients had more severe conditions and a higher mortality risk than NOIDD patients. It was common that IDD patients had a higher risk of neoplastic disease [17,21,26], and the malignant neoplasm was a major cause of death when patients had IDDs [20,27]. This phenomenon means that malignancy patients with immunodeficiencies have a worse prognosis. 

Additionally, there were more low-household-income families (0–25th percentile) in the group of malignancy patients with IDDs. In contrast to the general population, they had higher healthcare costs and resorted to Medicaid from the government, which further imposed a heavy economic burden on their families and on the national healthcare system. Almost one fifth of Medicare beneficiaries who had been discharged from a hospital had a 30-day readmission, which costed the US healthcare system 17.6 billion dollars annually [28].

However, no such information is currently available on the readmission of malignancy patients with IDDs, and our study appeared to fill that gap. Ryan et al. calculated that the readmission rate of the usual care was 33.8% at 30 days in 390 patients with advanced cancer [29]. In our study, the 30-day readmission rates of IDD and NOIDD patients were lower because the computational methods were different and because only the first 30 days of readmission were taken into account in this study. However, the readmission rate of the IDD patients was higher than that of the control patients in all the distributions of the 30-day intervals in our study, especially the first 30-day interval. That is, IDD patients with malignancies were more susceptible to readmission to the hospital, especially to UPR within the first 30 days after discharge, which was an indicator of their prognosis [30]. Our findings illustrated the temporality with the improved management and transitional care of tumor patients with IDDs after discharge. 

Compared with NOIDD, we found that the risk of 30-day, 90-day and 180-day readmissions was increased 1.32 fold, 1.27 fold and 1.28 fold in IDD patients with cancer, respectively. Our results suggested that IDDs were an independent risk factor for readmission in patients with malignancies in all periods after discharge. Consistent with our results, the prior review also showed that patients with HIV-1-associated malignancies had a higher mortality, which was an indicator of long-term prognoses that were different from 30-day UPR [19]. Additionally, a previous study showed that there was a 6-fold increase in the cancer mortality of CVID patients, especially in those with stomach cancer (an approximately 40-fold increase) [16]. CVID displayed a phenotype of impaired terminal B-cell differentiation and defective antibody responses, and CVID patients were susceptible to bacterial infections in the respiratory and gastrointestinal tracts [14,31,32]. Recent studies have pointed to the expansion of exhausted CD8 T cells (TEX) in CVID patients with complications and to the TEX prolonged antigen stimulation in the chronic inflammatory response that occurs by releasing inhibitory immune signals [33]. In a South African study, bacterial pneumonia was a major cause for hospital admissions among HIV patients [34]. Our results showed the same points that IDD patients were more susceptible to bacterial infections and that they suffered from pneumonia within 30 days after discharge. Therefore, malignant tumor patients with IDDs would have a poor prognosis both in the short term and the long term. 

In order to provide precise treatment, we evaluated that the readmission rate gradually decreased over time and that the majority of patients were not planned to be readmitted within 30-day. Therefore, we focused on the 30-day UPRs and found different underlying readmission risks in special populations. Similar to the other findings, we found that some specific populations had a higher 30-day UPR risk but not other malignancies, such as colorectal cancer by 1.32 fold, lung cancer by 1.23 fold, NHL by 1.16 fold, prostate cancer by 1.45 fold or stomach cancer by 2.34 fold. As a previous study suggested, stomach cancer had a 40-fold increase in cancer mortality, and the other cancers also had a high cancer mortality, which may not completely explain our results because we mainly paid attention to the short-term prognosis, not the overall survival [16]. Interestingly, IDDs may be a protective factor in patients with ovarian cancer with respect to 30-day UPR. Unfortunately, the exact mechanisms for why IDDs were an independent factor for the readmission of these tumors are unknown. Therefore, more investigations on the prognosis mechanisms are needed. 

Different from the general tumor patients, we also found that anemia was more common in patients with five malignancies and with immunodeficiencies. Anemia, a common hematologic complication of HIV-infected patients, was associated with decreased survival [35,36]. Cytopenia (anemia, neutropenia, thrombocytopenia, etc.) may be a typical first symptom of such an immunodeficiency, and it is particularly common in patients with antibody defects, CVID and selective immunoglobulin A deficiency [37,38]. Additionally, anemia is common in cancer patients [39]. A European prospective survey showed that the prevalence of anemia among cancer patients at enrolment was 39.3%, rising to 67% over a 6-month observation period [40]. The above information may be the cause of the higher incidence of anemia in IDD patients with malignancies within 30 days after discharge.

The finding of our study would help us to pay more attention to special populations and to develop target strategies to prevent complications in the treatment of malignancy patients with IDDs. IDDs should be treated as chronic conditions, and more attention should be paid to the prevention of anemia, opportunistic infections and pneumonia [14,19] as well as to the reduction of the possibility of tumor-related complications during treatment. Traditionally, anemia has been treated with blood transfusions, whereas transfusions were associated with an increased risk of dying in HIV-infected patients [35]. Therapy with recombinant human erythropoietin(r-HuEPO) has been shown to be well tolerated in anemia induced by HIV-infections, neoplasms and chronic diseases [35,36], and the r-HuEPO has been proven to elevate the hematocrit values and to reduce the transfusion requirements in HIV-infected patients who have endogenous erythropoietin levels of ≤500 IU/L [41]. However, there are complicated causes of anemia in primary immunodeficiency patients, which contain autoimmune hemolytic anemia [42], pernicious anemia [43], anemia of inflammation [44], and their usual treatment is treatment with oral dexamethasone, prednisone, IV steroids or IV rituximab42 [45]; the replacement of vitamin B12 [43]; and the combination of iron therapy and r-HuEPO [44], respectively. Transfusing irradiated red blood cells is recommended for immunodeficiency patients with severe anemia in order to alleviate acute symptoms [36,46]. As for preventing bacterial infections and pneumonia, immunoglobulins can significantly reduce the incidence of bacterial infections and prolong patient survival as well as reduce the frequency of autoimmune cytopenia [42]. Antibiotics are needed for acute treatment, and, in many cases, they are also recommended on a chronic basis as prophylaxis [42]. 

However, several limitations in our study should be noted when interpreting our results. First, the study population that we included were the 16 most common malignancies, and IDDs may be a potential risk factor in other cancers. Second, this study is limited by a retrospective cohort study design. Even though we adjusted for potential confounding factors, it is possible that unmeasured or unconsidered factors could affect the results.

## 5. Conclusions

In summary, we identified that immunodeficiencies were independently both short-term and long-term prognosis risk factors for 16 types of malignant tumors, especially the 30-day UPR, which brought heavy economic burdens on their families and on the national healthcare system. Anemia, bacterial infections and pneumonia, were the differential 30-day UPR causes in malignancy patients with immunodeficiencies, and they should be considered in their plan of discharge in order to improve their prognosis. Future studies focusing on the identification of novel screening guidelines for the readmission of neoplasm patients with IDDs are warranted.

## Figures and Tables

**Figure 1 cancers-15-00088-f001:**
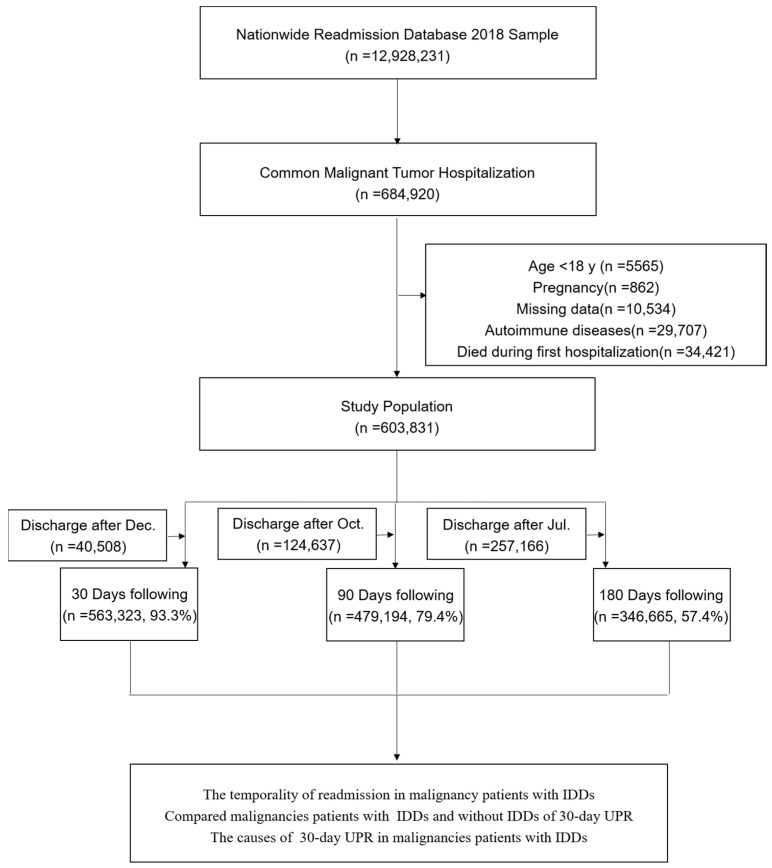
Inclusion and exclusion criteria.

**Figure 2 cancers-15-00088-f002:**
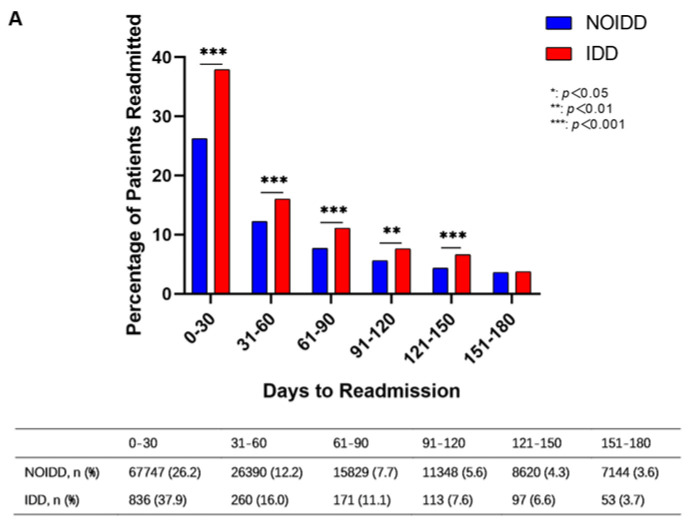
Malignancy patients’ (with IDDs and without IDDs (NOIDDs)) distribution of readmission to hospital during the 180-day follow-up. (**A**) NOIDD and IDD distribution of readmission during the 180-day follow-up; (**B**) PR and UPR distribution of the readmission of malignant tumor patients with IDDs during the 180-day follow-up.

**Figure 3 cancers-15-00088-f003:**
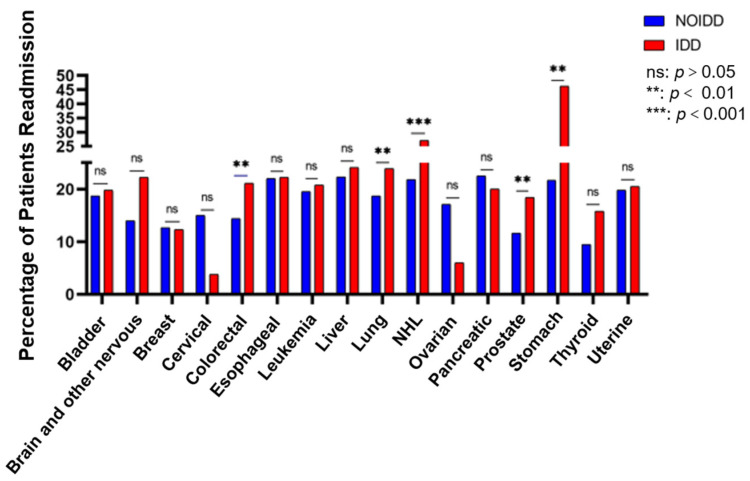
Malignancy patients with IDDs compared with those with NOIDDs of the 30-day unplanned readmission group.

**Figure 4 cancers-15-00088-f004:**
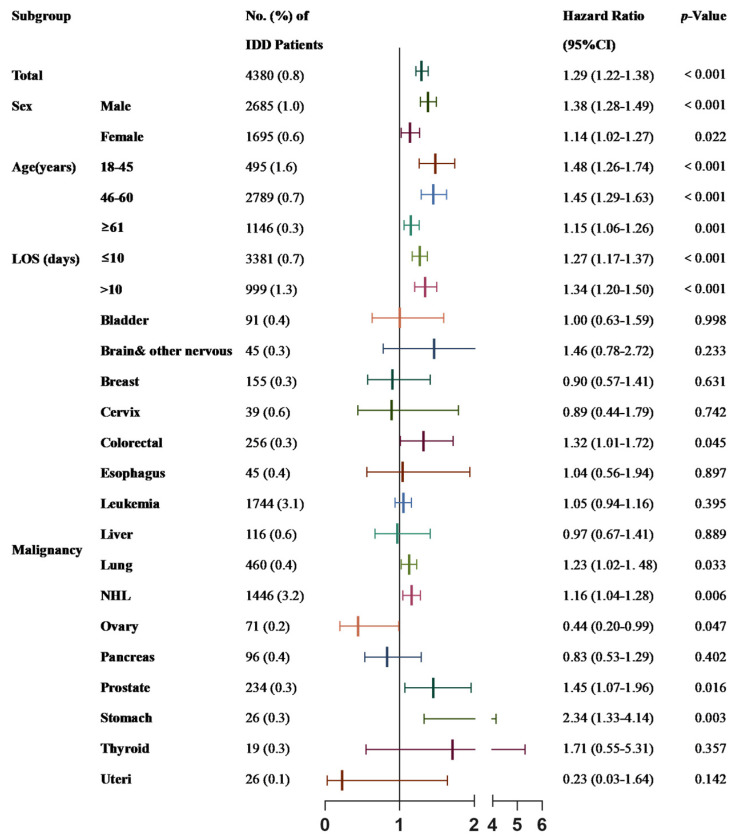
Subgroup analysis of 30-day unplanned readmission group (NOIDD ref. = 1). Analyses were adjusted for age, sex, body mass index, hypertension, diabetes, hyperlipidemia, severe hospitalization, location of residence, insurance type and household income.

**Figure 5 cancers-15-00088-f005:**
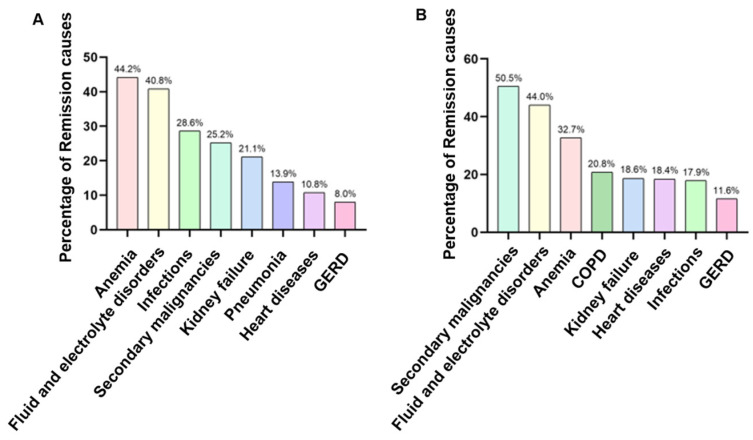
The 8 most common (tumor- and nontumor-related) causes of 30-day UPRs in colorectal cancer, lung cancer, non-Hodgkin lymphoma, prostate cancer and stomach cancer patients with IDDs (**A**) or without IDDs (**B**) (based on ICD-10-CM codes).

**Table 1 cancers-15-00088-t001:** Baseline characteristics of study population from the NRD that was classified as malignant tumor patients with or without immunodeficiency diseases.

Variables	IDD	NOIDD	*p*-Value
No. of cases, n	4852	598,979	
Male, n (%)	2993 (61.7)	304,906 (50.9)	*<*0.001
Age(years), n (%)			*<*0.001
1. 18–45	595 (12.3)	34,847 (5.8)	
2. 46–60	1231 (25.4)	125,740 (21.0)	
3. ≥61	3026 (62.4)	438,392 (73.2)	
Hypertension, n (%)	2620 (54.0)	364,731 (60.9)	*<*0.001
Diabetes, n (%)	1232 (25.4)	164,874 (27.5)	0.001
Hyperlipidemia, n (%)	1541 (31.8)	222,625 (37.2)	*<*0.001
BMI ≥ 25.0, n (%)	381 (7.9)	65,897 (11.0)	*<*0.001
Illness severity, n (%)			*<*0.001
1. Unclassified, minor and moderate	972 (20.0)	276,825 (46.2)	
2. Major and extreme	3880 (80.0)	322,154 (53.8)	
Mortality risk, n (%)			*<*0.001
1. Unclassified, minor and moderate	1894 (39.0)	314,526 (52.5)	
2. Major and extreme	2958 (61.0)	284,453 (47.5)	
LOS (days), n (%)			*<*0.001
1. ≤10	4742 (77.1)	517,390 (86.4)	
2. >10	1110 (22.9)	81,589 (13.6)	
Total charges, mean (SE)	114,912 (219,292)	81,344 (116,646)	*<*0.001
Patient location, n (%)			*<*0.001
1. “Central” counties with population ≥ 1 million	1580 (32.6)	170,012 (28.4)	
2. “Fringe” counties with population ≥ 1 million	1371 (28.3)	162,637 (27.2)	
3. Population of 250,000–999,999	1002 (20.7)	126,260 (21.1)	
4. Population of 50,000–249,999	402 (8.3)	55,841 (9.3)	
5. Micropolitan counties	292 (6.0)	46,925 (7.8)	
6. Not metropolitan or micropolitan counties	205 (4.2)	37,304 (6.2)	
Primary expected payer, n (%)			*<*0.001
1. Medicare	2932 (60.4)	367,355 (61.3)	
2. Medicaid	615 (12.7)	52,496 (8.8)	
3. Private insurance	1133 (23.4)	155,192 (25.9)	
4. Self-pay	91 (1.9)	9048 (1.5)	
5. No charge	13 (0.3)	1248 (0.2)	
6. Others	68 (1.4)	13,640 (2.3)	
Median household income, n (%)			0.001
1. 0–25th percentile (USD 1–USD 45,999)	1224 (25.2)	141,811 (23.7)	
2. 26th to 50th percentile (USD 46,000–USD 58,999)	1208 (24.9)	159,527 (26.6)	
3. 51st to 75th percentile (USD 59,000–USD 78,999)	1187 (24.5)	154,022 (25.7)	
4. 76th to 100th percentile (USD 79,000 or more)	1233 (25.4)	143,619 (24.0)	
Disposition of patient, n (%)			*<*0.001
1. Routine	3006 (62.0)	355,505 (59.4)	
2. Transfer to short-term hospital	60 (1.2)	6758 (1.1)	
3. Transfer to other	627 (12.9)	87,225 (14.6)	
4. Home health care	1106 (22.8)	145,154 (24.2)	
5. Against medical advice	50 (1.0)	3992 (0.7)	
6. Discharged alive, destination unknown	3 (0.1)	345 (0.1)	
Readmission (unadjusted), n (%)	2293 (47.3)	219,155 (36.6)	*<*0.001
Index admission by malignancy			
1. Bladder	100 (2.1)	27,644 (4.6)	*<*0.001
2. Brain and other nervous	50 (1.0)	16,820 (2.8)	*<*0.001
3. Breast	178 (3.7)	62,055 (10.4)	*<*0.001
4. Cervical	40 (0.8)	6930 (1.2)	0.031
5. Colorectal	282 (5.8)	80,937 (13.5)	*<*0.001
6. Esophageal	49 (1.0)	11,772 (2.0)	*<*0.001
7. Leukemia	1879 (38.7)	60,180 (10.0)	*<*0.001
8. Liver	126 (2.6)	20,883 (3.5)	0.001
9. Lung	494 (10.2)	117,233 (19.6)	*<*0.001
10. non-Hodgkin lymphoma	1683 (34.7)	49,378 (8.3)	*<*0.001
11. Ovarian	82 (1.7)	36,371 (6.1)	*<*0.001
12. Pancreatic	101 (2.1)	29,573 (4.9)	*<*0.001
13. Prostate	252 (5.2)	78,727 (13.1)	*<*0.001
14. Stomach	30 (0.6)	11,417 (1.9)	*<*0.001
15. Thyroid	20 (0.4)	7291 (1.2)	*<*0.001
16. Uterine	30 (0.6)	19,585 (3.3)	*<*0.001

**Table 2 cancers-15-00088-t002:** Adjusted hazard ratio of the association of IDDs with readmission stratified by 30 days, 90 days and 180 days.

Elective	Immunity Status	30-Day Adjusted * HR (95% CI)	*p*-Value	90-Day Adjusted * HR (95% CI)	*p*-Value	180-Day Adjusted * HR (95% CI)	*p*-Value
Uncategorized	NOIDD	1.0 (ref)		1.0 (ref)		1.0 (ref)	
	IDD	1.32 (1.25–1.40)	<0.001	1.27 (1.21–1.34)	<0.001	1.28 (1.22–1.35)	<0.001
PR	NOIDD	1.0 (ref)		1.0 (ref)		1.0 (ref)	
	IDD	1.68 (1.46–1.92)	<0.001	1.38 (1.22–1.57)	<0.001	1.33 (1.17–1.52)	<0.001
UPR	NOIDD	1.0 (ref)		1.0 (ref)		1.0 (ref)	
	IDD	1.29 (1.22–1.38)	<0.001	1.27 (1.21–1.34)	<0.001	1.29 (1.22–1.37)	<0.001

HR represents hazard ratio; CI represents confidence interval. * Analyses adjusted for age, sex, body mass index, hypertension, diabetes, hyperlipidemia, LOS, location of residence, insurance type and household income.

**Table 3 cancers-15-00088-t003:** Difference analysis of the 30-day unplanned readmission causes in IDD and NOIDD patients.

Causes of UPR, n (%)	IDD, n = 611	NOIDD, n = 50,722	*p*-Value
Anemia	270 (44.2)	16,588 (32.7)	<0.001
Fluid and electrolyte disorders	249 (40.8)	22,298 (44.0)	0.112
Infections	175 (28.6)	9099 (17.9)	<0.001
Secondary malignancies	154 (25.2)	25,606 (50.5)	<0.001
Kidney failure	129 (21.1)	9444 (18.6)	0.116
Pneumonia	85 (13.9)	5397 (10.6)	0.009
Heart diseases	66 (10.8)	9337 (18.4)	<0.001
GERD	49 (8.0)	5882 (11.6)	0.006

## Data Availability

The data presented in this study are available upon reasonable request.

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
