# Peer review of "Immunodeficiencies Push Readmissions in Malignant Tumor Patients: A Retrospective Cohort Study Based on the Nationwide Readmission Database"

_cancers, 2022, doi:10.3390/cancers15010088_

Round 1
Reviewer 1 Report
Wang et al., discussed the pressing issue of evaluating the hospital readmission rates in patients with cancer and immunodeficiency in the article, "Immunodeficiencies Push Readmissions in Malignant Tumor 2 Patients: A Retrospective Cohort Study Based on Nationwide 3 Readmission Database". The authors should be commended for their work. I have a few suggestions for the authors
- Page 4, line 107: Please replace the word such as with, "that include"
- Methods: Can authors detail why the year 2018 was selected instead of 2019 or 2021.
- Methods: Can authors clarify that the patients in IDD cohort has only primary immunodeficiency syndromes rather than acquired immunodeficiency stemming from leukopenia from cancer-directed therapy?
- Methods: In general, it is expected that patients with immunodeficiency states have higher re-admission rates. Wondering if authors compared the re-admission rates in patients with immunodeficiencies withOUT cancer and patients with immunodeficiencies and cancer.
Author Response
Point-by-Point Response to Reviews
Dear Editors and Reviewers:
Thank you for your letter and for the reviewers’ comments concerning our manuscript entitled “Immunodeficiencies Push Readmissions in Malignant Tumor Patients: A Retrospective Cohort Study Based on Nationwide Readmission Database” (ID: cancers-2078696). Those comments are all valuable and very helpful for revising and improving our paper, as well as the important guiding significance to our researches. We have studied comments carefully and have revised our manuscript accordingly.
The track changes mode was used in the revised version. A clear version was also provided. Revised portion are marked in red in the paper. The main corrections in the paper and the responds to the reviewer’s comments are as following:
Responds to the reviewer’s comments:
Reviewers' comments:
Reviewer #1: In this manuscript authors demonstrate that immunodeficiencies push readmissions in malignant tumor 2 patients: a retrospective cohort study based on nationwide readmission database. First, authors needed to revise some words and replace the “such as” with “that include”. Furthermore, authors needed to detail why the year 2018 was selected instead of 2019 or 2021. In addition, authors needed to clarify that the patients in IDD cohort have only primary immunodeficiency syndromes rather than acquired immunodeficiency stemming from leukopenia from cancer-directed therapy. Last, comparing the readmission rates in patients only with immunodeficiencies and patients with immunodeficiencies and cancer.
Response:
Thanks a lot for your insightful comments. We found that immunodeficiencies are independently associated with higher readmission risks of colorectal cancer, lung cancer, non-Hodgkin lymphoma, prostate cancer and stomach cancer. In order to detect patients at high risk of early readmission and reduce the burden of society, strategies should be considered to prevent the causes of readmission as post-discharge plan in this population. Depending on your suggestions, we have made the following corrections in the text.
1) Authors needed to revise some words and replace the “such as” with “that include” in Page 4, line 107.
Response:
Thanks for your suggestions. We have replaced the “such as” with “that include” in Page 4, line 108.
2) Methods: Can authors detail the reason why the year 2018 was selected instead of 2019 or 2021.
Response:
We agree that this would be very interesting to study. Nationwide Readmissions Database (NRD), the part of the Healthcare Cost and Utilization Project (HCUP), supports various types of analyses of national readmission rates for all patients regardless of the expected payer for the hospital stay of the 28 states in an individual year. Based on the NRD unique record identifier, it allows to track of patients between hospitals across states, but not across years. NRD included many diseases every year, and NRD 2018 has the same equal effect in approving our views compared with the other year’s NRD.
3) Can authors clarify that the patients in IDD cohort have only primary immunodeficiency syndromes rather than acquired immunodeficiency stemming from leukopenia from cancer-directed therapy?
Response: Thank you for your suggestions. In our study, immunodeficiency diseases included primary immunodeficiency diseases and secondary immunodeficiency diseases(Supplemental Table 1).The leukopenia from cancer-directed therapy was not included in our study.
4) In general, it is expected that patients with immunodeficiency states have higher readmission rates. Wondering if authors compared the re-admission rates in patients with immunodeficiencies without cancer and patients with immunodeficiencies and cancer.
We agree that it is a very useful to study. In this study, we mainly focused on immunodeficiencies in tumor patients. We included patients with the 16 common malignant tumors and identified immunodeficiencies in these patients. However, immunodeficiencies are rare diseases, we will conduct a study to focus this population to find the relationship between immunodeficiencies and readmission.

Reviewer 2 Report
Perspective:
- The introduction gave sufficient background and referenced properly. Their primary objective aimed to correlate immunodeficiencies (IDDs) with short and long readmission risk. The statistical analysis focused on Cox regression for 30-, 90- and 180-day UPRs. They concluded that there was a statistically significance between IDD vs NOIDD.
Their secondary objective was to understand which stage of disease is the most common for readmissions after 180days.
- Materials and Methods: Might be beneficial to add the limitations of the study that were only mentioned in line 345. The limitations include: study population, cohort study design and possible unconsidered factors.
- The presentation of the data was clear and figures were relevant to the study.
- The flow of the paper is appropriate and there were no grammatical errors detected.
Additional Comments:
- The introduction: May need correction on line 71. “prevalence can range from 1/500 to 1/500,00014”
- Supplementary materials show “404 error”. Would be beneficial to have the SPSS model for further reference.
- You identified 30-days as the most common timeframe for planned readmissions, and you henceforth further did subgroup analyses (section 3.4) and causes (section 3.5), but regarding unplanned readmissions (UPR), according to Table 2, the HR at 30-days was almost identical to 180-days. So why not further investigate subgroup and causes analyses (like for 30day PR in section 3.4 and 3.5)?
Author Response
Point-by-Point Response to Reviews
Dear Editors and Reviewers:
Thank you for your letter and for the reviewers’ comments concerning our manuscript entitled “Immunodeficiencies Push Readmissions in Malignant Tumor Patients: A Retrospective Cohort Study Based on Nationwide Readmission Database” (ID: cancers-2078696). Those comments are all valuable and very helpful for revising and improving our paper, as well as the important guiding significance to our researches. We have studied comments carefully and have revised our manuscript accordingly.
The track changes mode was used in the revised version. A clear version was also provided. Revised portion are marked in red in the paper. The main corrections in the paper and the responds to the reviewer’s comments are as following:
Responds to the reviewer’s comments:
Reviewers' comments:
Reviewer #2: In the manuscript, " Immunodeficiencies Push Readmissions in Malignant Tumor Patients: A Retrospective Cohort Study Based on Nationwide Readmission Database " the authors gave sufficient background and referenced properly. Their primary objective aimed to correlate immunodeficiencies (IDDs) with short and long readmission risk. The statistical analysis focused on Cox regression for 30-, 90- and 180-day UPRs. They concluded that there was a statistically significance between IDD vs NOIDD. Their secondary objective was to understand which stage of disease is the most common for readmissions after 180days.
Response:
Thank you very much for your kind comments. our study is unique because, to date, no study has assessed the impact of immunodeficiency on short- and long-term clinical outcomes of patients with malignant tumors, which allowed for calibration of readmission risk stratification in these population. To address this important knowledge gap, we analyzed the NRD database, a large contemporary nationwide database in the USA with a broad spectrum of data on physical characteristics and immune status, and then evaluated the association between immunodeficiency and readmission risk of tumor patients to provide a reference for clinical prevention and intervention. We believe these findings will be of great interest to readers generally, and particularly to researchers working on tumor patients with immunodeficiency and related areas. Depending on your suggestions, we have made the following corrections in the text.
Review of this manuscript generated the following comments and questions for the authors:
- The introduction: May need correction on line 71. “prevalence can range from 1/500 to 1/500,00014”
Response:
Thank you for your suggestions. “prevalence can range from 1/500 to 1/500,00014”. 14th is one of the References. We have made the following corrections in the text.
- Supplementary materials show “404 error”. Would be beneficial to have the SPSS model for further reference.
Response:
Thanks for your suggestions. We have uploaded the supplementary materials again in the text.
- You identified 30-days as the most common timeframe for planned readmissions, and you henceforth further did subgroup analyses (section 3.4) and causes (section 3.5), but regarding unplanned readmissions (UPR), according to Table 2, the HR at 30-days was almost identical to 180-days. So why not further investigate subgroup and causes analyses (like for 30day PR in section 3.4 and 3.5)?
Response:
Thanks for your suggestions. We focus on early readmission (30day readmission), especial the unplanned readmission because of many tumor patients have to go through surgery. The early prognosis has great meaning for tumor patients. Therefore, we especially investigate early readmission subgroup and causes of planned and unplanned readmission in order to make post-discharge strategies for this population.
Finally, we tried our best to improve the manuscript and made some changes in the manuscript. These changes will not influence the content and framework of the paper. And here we listed the changes and marked in blue in revised paper.
We appreciate for Editors/Reviewers’ warm work earnestly, and hope that the correction will meet with approval.
Once again, thank you very much for your comments and suggestions.

Round 2
Reviewer 1 Report
The authors addressed all the comments appropriately.